# Measurement Accuracy of the HTC VIVE Tracker 3.0 Compared to Vicon System for Generating Valid Positional Feedback in Virtual Reality

**DOI:** 10.3390/s23177371

**Published:** 2023-08-24

**Authors:** Sebastian Merker, Stefan Pastel, Dan Bürger, Alexander Schwadtke, Kerstin Witte

**Affiliations:** Sports Engineering and Movement Science, Otto-von-Guericke University, 39106 Magdeburg, Germany; sebastian.merker@ovgu.de (S.M.); dan.buerger@ovgu.de (D.B.); alexander.schwadtke@st.ovgu.de (A.S.); kerstin.witte@ovgu.de (K.W.)

**Keywords:** Vicon system, HTC VIVE Tracker, virtual reality, motion capture

## Abstract

For realistic and reliable full-body visualization in virtual reality, the HTC VIVE Tracker could be an alternative to highly complex and cost- and effort-intensive motion capture systems such as Vicon. Due to its lighter weight and smaller dimensions, the latest generation of trackers is proving to be very promising for capturing human movements. The aim of this study was to investigate the accuracy of the HTC VIVE Tracker 3.0 compared to the gold-standard Vicon for different arrangements of the base stations and various velocities during an athletic movement. Therefore, the position data from three trackers attached to the hip, knee and ankle of one sporty participant were recorded while riding a bicycle ergometer at different pedaling frequencies and different base station arrangements. As parameters for the measurement accuracy, the trajectories of the linear motion of the knee and the circular motion of the ankle were compared between VIVE and Vicon by calculating the spatial distance from the raw data at each point in time. Both the pedaling frequency and the arrangement of the base stations significantly affected the measurement accuracy, with the lowest pedaling frequency of 80 rpm and the rectangular arrangement recommended by the manufacturer showing the smallest spatial differences of 10.4 mm ± 4.5 mm at the knee and 11.3 mm ± 5.1 mm at the ankle. As the pedaling frequency increased gradually (120 rpm and 160 rpm), the measurement accuracy of the trackers per step decreased less at the knee (approximately 5 mm) than at the ankle (approximately 10 mm). In conclusion, the measurement accuracy for various athletic skills was high enough to enable the visualization of body limbs or the entire body using inverse kinematics in VR on the one hand and, on the other hand, to provide initial insights into the quality of certain techniques at lower speeds in sports science research. However, the VIVE trackers are not suitable for exact biomechanical analyses.

## 1. Introduction

In recent years, technological progress has enabled the development of learning scenarios using hard- and software systems that were not initially associated with an educational purpose. For example, immersive virtual reality (VR) applications mainly used for entertainment are now considered for research and learning tools due to their high potential in facilitating previously unrealizable learning scenarios within real-world settings. Therefore, diverse VR applications have been integrated into different research areas, such as medicine [1], psychology [2], engineering [3], sports [4,5] and more.

For each listed example, the question arises of how VR could be used to support learning [6] or practice skills to reduce the error probability in real-world scenarios. Hereby, it is essential to recognize the setup of each VR system, which could narrow down or amplify possible applications. In general, an immersive VR system contains several components, such as the head-mounted display (HMD), base stations and controller. Those can differ depending on the manufacturer and the applications. A remaining challenge with today’s VR application is to enable a realistic-looking, reliable full-body visualization. To realize this, the usage of a highly complex and costly motion-capturing system, such as Vicon Shogun (Oxford, UK), is necessary to synchronize users’ movements to the avatar’s skeleton [7]. This passive marker system consists of a list of space-consuming infrared cameras (often more than 10), tripods, an advanced computer system (SSDs, powerful graphic cards, etc.) and more. Many studies have already used those techniques to facilitate full-body visualization within the laboratory setup [4,8]. Considering VR as a learning tool that may also appeal for private use, such as home-based training, it would be unrealistic to assume that the previously described technology is affordable for a private user or household. Therefore, other systems emerged as a practical alternative [9,10]. High Tech Computer Corporation (HTC, Taiwan) has developed a wireless, battery-powered photoreceptive tracker (the HTC VIVE Tracker), allowing for the visualization of body limbs through replacing the trackers’ mesh with a realistic-looking body mesh (often the forearms) [11]. The position of the tracker is determined via the dead reckoning of an integrated 9-DOF IMU and corrected using 18 infrared sensors mounted at different angles, which interact with the base stations to avoid the typical drift of the IMU [12,13]. These trackers could be an alternative to such systems realizing the visualization of the user’s body [14] that cost only approximately one percent compared to the full-featured Vicon system. By placing the tracker on the user’s body, the three-dimensional position over time is captured, and the impression of having control of the virtual body is fulfilled. Additionally, the trackers can be used not only for visualization, but also for collecting user’s movement data to expand the feedback on their movement patterns. This may improve the impact of VR training, especially within the sports sector, in which individualized feedback is essential.

Before confirming the validity and reliability of the tracker usage, it must be compared with the already established measuring tools. Therefore, this paper aims to compare the HTC VIVE Tracker 3.0 with the Vicon Nexus system, which is also known as the gold-standard [15], evaluated through a high measurement precision of less than 2 mm [16]. Other systems, such as Kinect [17] or wi-GAT [18], have already been evaluated by comparing their data with the outcome from Vicon. Much work has already been performed concerning validating previous models of the VIVE tracker compared to Vicon [7] or detecting movement trajectories [19]. Van der Veen et al. (2019) examined the accuracy of the position from the second-generation VIVE trackers compared to an optoelectronic 3D motion-capturing system by measuring movements or reaches conducted by a robot executing more simple movements and humans playing a VR game. The authors stated an acceptable accuracy for translations (averaged deviation under 1 cm) and rotations (averaged rotation shifting of approximately 1.6°). They also concluded a high practical value enabled by the trackers and considered the increased tracking space provided by the next generation of base stations [7]. Nevertheless, the authors emphasized the importance of examining future versions of VR motion-tracking devices, especially when higher movement velocities need to be tracked. Borges et al. (2018) also tested the trackers for the Astrobee robot. They developed a new open-source tracking algorithm that differs from the source code of VIVE, leading to higher accuracy up to two orders of magnitude [20]. The authors stated the high precision of the VIVE trackers when static measurements took place, but during motion, the accuracy decreased from the millimeter to the meter range [20]. Caserman et al. (2019) used the trackers to animate the avatar’s motion, being able to track joint rotations and positions. The authors stated a low end-to-end latency of less than seven milliseconds [11].

Regarding the current state of the art, the practical use of the VIVE trackers was mainly determined by analyzing movements conducted by robots under laboratory settings. Additionally, those movements are hard to compare to more sport-specific ones, in which fast and partially unpredictable movements often occur. Concerning the technical component, third-generation VIVE trackers (3.0) appeared on the market, being more user-friendly due to the smaller size and weight. Therefore, the goal of the current study is to evaluate the measurement accuracy of the current version of VIVE trackers by comparing them with the Vicon system for higher velocities and accelerations during an athletic movement. For this purpose, we capture the positional data of the hip, knee and ankle while the participant rides a bicycle ergometer at various pedaling frequencies. Furthermore, we vary the placement of the base stations to determine if tracking accuracy increases with a larger number of base stations with an unrestricted line-of-sight and closer distance to the trackers. As an indicator of the measurement accuracy, we compare the recorded trajectories at the knee and ankle of both systems. The translational motion of the knee in cycling is characterized by high speeds and even higher accelerations due to the direct reversal of motion in each cycle. In contrast, the ankle moves circularly around an almost fixed axis of rotation at a constant angular velocity and radial acceleration. We chose these two measurement points for the comparison between VIVE and Vicon because the uniform and circular movements described are characteristic of many other athletic movements. In addition, we calculate the radius of the ankle motion to find out whether the VIVE trackers tend to overestimate or underestimate spatial distances compared to Vicon.

## 2. Materials and Methods

### 2.1. Experimental Setup

For the tracking validation of the VIVE Tracker 3.0, the movement data of three trackers were compared with the movement data measured with a Vicon system. For this purpose, as shown in Figure 1, one 14 mm Vicon marker was attached to each tracker. While the coordinate origin of the VIVE tracker is in the mounting thread at the bottom, the origin of the spherical Vicon marker is in its center. Due to this difference in origin, the orientational information from the trackers was used to determine which axis needed to be adjusted regarding the position of the Vicon marker. As can be seen on the right side in Figure 1, the difference between the coordinate origins of both measuring systems was 30 mm. Accordingly, the recorded coordinates of the VIVE tracker were corrected using this offset to allow for a comparison of the trajectories of both systems. This correction was applied directly in the Unity scene by adding an empty game object defined as a child, which adjusted the translations and rotations.

The measurement took place in a laboratory with a preinstalled Vicon system. This system consisted of 13 infrared cameras (4 Vantage V5, 9 Vero v2.2) and the software Vicon Nexus (version 2.12.1). The movement data of the three trackers were collected wirelessly using one VIVE Dongle each, four HTC VIVE Base Station 2.0 base stations (see Figure 2), the HTC VIVE Pro Eye, the software Unity (2021.25f1) and SteamVR (1.23). The Unity project was kept as simple as possible by using only three objects, which moved according to the trackers using the “SteamVR_TrackedObject” script. The measurement PC had a seventh-generation Intel Core i7 processor, 16 GB DDR4 RAM, 256 GB SSD and an NVIDIA GeForce GTX 1080 graphics card with 8 GB memory. The trackers were first calibrated and then attached to the participant’s hip, knee and ankle (see Figure 3). In the middle of the measurement volume was a bicycle ergometer. Movement data were collected during cycling and exported using a self-written C# script.

### 2.2. Procedure

One sporty participant (25 years, male) completed the task, and his data were collected. The task was to ride the bike after the start signal given by the experimenter. The influence of three factors on the accuracy of the trackers’ position tracking was examined. One factor was the frequency in rounds per minute (rpm) that the participant pedaled with, which altered between 80 rpm, 120 rpm and 160 rpm. The factor was chosen to examine the influence of the movement speed and the associated acceleration on the tracking accuracy. With the help of a metronome, the participant was able to maintain the required frequency. In addition, he saw his current frequency on the ergometer and pedaled as close as possible to the demanded speed. Another factor was the position of the base stations, which were either placed in a rectangle (5 m × 5 m) or a kite formation (see Figure 2). The manufacturer recommended a rectangle formation. The kite formation was chosen to reduce the distance between the trackers and the base stations and to allow for three base stations to have immediate visual contact with the attached trackers to increase accuracy.

In total, there were six possible combinations, as shown in Figure 4, in their conducted order. Every condition was performed thrice for 30 s, one directly after the other, with a break of approximately 30 s. Afterward, there was a break of approximately 60 s, during which the condition was altered. After three conditions, there was a five-minute break to change and calibrate the base stations’ positions and minimize the participant’s exhaustion. To synchronize both measurement systems in time, the participant moved his knee to the side at the beginning of a trial. The resistance of the ergometer was set to 100 watts. For comparing both systems, the recording frequency of both systems was set to their optimum (Vicon 200 Hz; VIVE 90 Hz).

### 2.3. Data and Statistical Analysis

To assess how well the captured movement of the VIVE trackers compares to the gold-standard Vicon for analyzing the fast athletic movements, we focused primarily on analyzing the spatial distances at any point in time of the captured trajectories from the markers at the knee and ankle. We also computed the radius of the ankle markers (see Figure 3) for each measurement system to be able to determine statements about the direction of the deviations.

Based on the described knee movement to the side at the start of each trial, the two measurement systems were synchronized in time by removing all measured values before this event. Then, the hip-to-ankle distances were calculated for all time points to identify each cycle, with a single cycle always starting with a minimum hip-to-ankle distance. After determining the start time for the first cycle, the data set was trimmed according to the required number of complete cycles. To achieve the target measurement duration of 30 s, the number was exactly 40 cycles at a pedaling frequency of 80 rpm, 60 cycles at 120 rpm and 80 cycles at 160 rpm. Due to the different axis orientations of the two measurement systems, we used their corresponding unit vectors to calculate a transformation matrix to synchronize the spatial orientation of the coordinate systems. Therefore, the center of the hip marker trajectory was set as the origin, while the corresponding vectors to the knee and ankle spread the x–y plane, and the orthogonal vector through the origin represented the *z*-axis. The following figure (Figure 5) shows an example of the synchronized trajectories.

An initial comparison of the trimmed distance–time series between VIVE and the gold-standard Vicon revealed that the actual recording frequency of the VIVE trackers differed from the 90 Hz frequency specified by the manufacturer and configured in Unity (see Figure 6). To determine the actual recording frequency, we extracted the time stamps of all minimum and maximum distances between the hip and ankle for both systems, as these occurred simultaneously in reality. Figure 7 shows the time difference between VIVE and Vicon of the related extreme values for one trial as a function of the time measured with Vicon. As can be seen, a linear relationship existed between the time difference of both systems and the measurement times. A linear regression was used to calculate a correction factor for each trial of the VIVE trackers’ time points from these data.

Using the respective correction factors, which corresponded to the reciprocal of the slope of the linear regression, the actual recording frequencies of the VIVE trackers were determined for all trials and their time stamps were adjusted. The figure below (Figure 8) shows the distance–time series of the corrected VIVE data compared to the original Vicon data. From this graph, the temporal correspondence of the cycles could be easily seen, so that with the help of the temporally corrected data, a comparison of both systems was now possible at any point in time.

To allow for a direct comparison of the trajectories of Vicon and VIVE at each measurement time, an interpolation of all measured values to an identical frequency had to be performed due to the different recording frequencies of both systems. We decided to use an interpolation frequency of 80 Hz due to the fluctuating actual recording frequency of the VIVE data. The interpolation was performed using the linear approximation. After synchronizing the raw data in time and space, correcting the VIVE timestamps and interpolating the data, we calculated the spatial distance between the VIVE tracker and Vicon marker for each measurement time point. To assess whether the VIVE trackers tended to overestimate or underestimate distances, for the near-circular motion of the ankle, we chose its radius as a measure of accuracy. We compared this radius for all trials between Vicon and VIVE. A frequency analysis of the time series was performed for each trial to ensure that the specified pedaling frequency was maintained.

First, the prerequisites of the statistical procedures were checked. Due to the sample size of approximately 2400 values per group and the fact that the ANOVA was very robust against non-normally distributed values, the statistical test of the normal distribution could be omitted in principle [21,22,23]. Nevertheless, for a better understanding of the data, a graphical analysis of the measured value distributions was performed. To determine the outliers, we chose to use the median absolute deviation since the mean and the standard deviation were strongly impacted by outliers [24]. Reliability testing of the VIVE trackers was performed over the three trials per condition by comparing trial 1 with trial 2 and trial 3, respectively. For this purpose, we calculated Pearson’s correlation coefficient for the mean distances grouped with pedaling frequency, base station position and tracker position.

To examine the previously calculated spatial distances at the knee and ankle, we used a two-factor ANOVA with Bonferroni-corrected post hoc tests for the distance as the dependent variable and the between-subject factor pedaling frequency (80 rpm, 120 rpm, 160 rpm) and base station position (rectangle, kite) for each tracker position. The inferential statistical analysis described was performed for each individual trial as well as on the combined data from all three trials. This ANOVA was used primarily to examine the significance of the main effects.

The analysis of the radii of the ankle tracker movement to determine in which direction the position data measured with VIVE deviated from Vicon was performed using a two-factor ANOVA with repeated measures and Bonferroni-corrected post hoc tests. The radius was analyzed as a dependent variable on the within-subject factor system (Vicon, VIVE), the between-subject factor pedaling frequency (80 rpm, 120 rpm, 160 rpm) and base station position (rectangle, kite). This ANOVA was mainly used to examine the interaction effects of both systems and the remaining factors. The effect sizes were estimated by using η_p_^2^ (0.01 = small effect; 0.06 = moderate effect; >0.14 = large effect). If no sphericity was given, the Greenhouse–Geisser correction was used to interpret the results.

For the data collection, analyses and visualization, R (version 4.1.2) and SPSS (version 29) were used.

## 3. Results

The analysis of the pedaling frequency was based on the distance–time series recorded with Vicon. For the specified frequencies of 80 rpm, 120 rpm and 160 rpm, the average frequencies were 79.8 rpm ± 0.4 rpm, 118.6 rpm ± 2.1 rpm and 157.4 rpm ± 3.0 rpm, respectively. Accordingly, it could be stated that the targeted movement speeds were achieved with sufficient accuracy. As described in the previous chapter, the recording frequencies of the VIVE trackers deviated from the configured recording frequency. Applying the correction, the actual recording frequency averaged over all trials was 86.9 Hz ± 1.2 Hz. Figure 9 shows the distribution of all measured values as a function of the various influencing factors. As could be clearly seen, the measured values were approximately normally distributed. Furthermore, it could already be clearly seen in Figure 9 that the distributions became flatter with increasing pedaling frequency on the one hand, which was equivalent to a larger standard deviation, and on the other hand, their peak increasingly shifted to the right, i.e., the spatial distances between both systems became larger.

Reliability testing revealed very high correlations between trials one and two (*r* = 0.934, *p* < 0.001) and trials one and three (*r* = 0.967, *p* < 0.001). To ensure that the analysis of the combined data from all three trials did not bias the results, the effect sizes of the factor pedaling frequency and base station position were first analyzed for each trial and tracker position and then compared. The analysis showed small-to-medium effects for the factor base station position (knee: 0.017 < *η_p_*^2^ < 0.096; ankle: 0.029 < *η_p_*^2^ < 0.120) and large effects for the factor pedaling frequency (knee: 0.254 < *η_p_*^2^ < 0.297; ankle: 0.435 < *η_p_*^2^ < 0.493). As a result of the high correlation and consistent effect sizes across the three trials, the measurement series of the trials were combined for all further analyses.

First, we focused on the evaluation of the spatial distances for the tracker at the knee, which moved almost linearly, as can be seen in Figure 5. As shown in Table 1, significant differences occurred for both main effects as well as the interaction. The pedaling frequency with *η_p_*^2^ = 0.275 had a large effect and the position of the base stations with *η_p_*^2^ = 0.048 had a small effect on the measurement accuracy of the VIVE trackers at the knee. From the pairwise comparisons, the rectangular base station position led to significantly smaller deviations between Vicon and VIVE at pedaling frequencies of 80 rpm and 160 rpm. At the frequency of 120 rpm, slightly smaller but still significant (*p* = 0.007) deviations could be seen for the kite-shaped arrangement of the base stations. The largest difference of 13.03 mm between the two base station positions was found for the pedaling frequency of 160 rpm. Regardless of the frequency, the deviations for the rectangle base station position were significantly smaller, by an average of 4.80 mm, compared to the kite arrangement. The analysis of the pairwise comparisons of the pedaling frequency showed that the measurement accuracy decreased significantly with increasing speed in each case (see Table 1). Here, the mean differences between 80 rpm and 120 rpm (Δ*d*_80 rpm → 120 rpm_ = 3.86 mm) were smaller than between 120 rpm and 160 rpm (Δ*d*_120 rpm → 160 rpm_ = 12.39 mm).

In addition to evaluating the nearly linear motion of the knee, we also focused on the circular motion of the ankle (see Figure 5). The results of the spatial distances between the VIVE tracker and Vicon marker at this position are shown in Table 2. Again, significant differences for the dependent variable distance could be seen for the two main effects and the interaction term. Compared to the tracker at the knee, the pedaling frequency had an even larger effect on the measurement accuracy with *η_p_*^2^ = 0.458 than the base station position with *η_p_*^2^ = 0.064. If we looked again at the pairwise comparisons for the arrangement of the base stations, we saw the same pattern as for the tracker at the knee. While at frequency levels of 80 rpm and 160 rpm the rectangular arrangement yielded significantly lower deviations (*p* < 0.001), the reverse was true at 120 rpm. Here, the measurements of the kite arrangement were again slightly but significantly more accurate than for the rectangular base station position. The largest difference between the groups of the arrangement was obtained with 13.66 mm at the highest pedaling frequency of 160 rpm. Overall, the rectangular base station position resulted in a significantly smaller deviation of 5.82 mm compared to the kite arrangement. As previously observed for the tracker at the knee, the deviations increased significantly with increasing pedaling frequency (see Table 2). Regardless of the base station position, the measurement accuracy between the pedaling frequencies of 80 rpm and 120 rpm also decreased less (Δ*d*_80 rpm → 120 rpm_ = 6.38 mm) than between the frequencies of 120 rpm and 160 rpm (Δ*d*_80 rpm → 120 rpm_ = 18.53 mm). In general, it could be determined for all pedaling frequencies depending on the base station positions that the deviations were greater for the tracker at the ankle than for the tracker at the knee.

Since the absolute value was always used for the calculation of the spatial distance, the information in which direction the two systems deviated from each other was lost. Therefore, a further analysis was performed for the radius of the approximately circular motion of the ankle to prove whether the VIVE trackers overestimated or underestimated the range of motion. For the statistical analysis of the radii, we used only the data from the first trial. The results of the descriptive and inferential statistical analysis are presented in Table 3. Here, it could be seen that all interactions with the within-subject factor system had a significant influence (*p* < 0.001). Furthermore, the pedaling frequency evoked significant differences for the radius (*p* < 0.001). Only the base station position factor had no significant influence (*p* = 0.165). As expected from the previous results, the within-subject factor system had the largest effect with *η_p_*^2^ = 0.736. In comparison, the between-subject factors of pedaling frequency and base station position had only a small effect (*η_p_*^2^ = 0.038) and no effect (*η_p_*^2^ < 0.001), respectively. From the descriptive results, the VIVE trackers continuously measured larger radii than Vicon across all measurement conditions. The radius measured with VIVE also increased with increasing rotational speed, while that of Vicon remained unchanged. While the differences of the radii between Vicon and VIVE were only 7.7 mm at 80 rpm, the difference grew to 25.6 mm for 160 rpm. The increase in the differences between the frequencies 80 rpm and 120 rpm with 7.5 mm was smaller than the increase between 120 rpm and 160 rpm with 10.4 mm.

The Bland–Altman diagrams confirmed the previously described trend of systematic measurement errors between VIVE and Vicon (see Figure 10). Pearson’s correlation test between VIVE and Vicon showed a large effect (*r* > 0.5) for all conditions. The largest effect (*r* = 0.996, *p* < 0.001) was found at the pedaling frequency of 80 rpm and the rectangular arrangement of the base stations. On the other hand, the smallest effect (*r* = 0.713, *p* < 0.001) was shown for the kite-shaped setting of the base stations at the pedaling frequency of 160 rpm. Moreover, it could be observed that the dispersion of the deviations between both measuring systems slightly depended on the height of the measured values in the investigated radius measuring range between 90 and 180 mm. The largest correlation (*r* = 0.570, *p* < 0.001) between the difference and mean existed for the pedaling frequency of 80 rpm and the rectangular base station arrangement. The lowest correlation (*r* = 0.134, *p* < 0.001) was observed for the same pedaling frequency of 80 rpm for the kite-shaped arrangement.

## 4. Discussion

The aim of this study was to investigate whether the VIVE trackers were usable for future VR training sessions or, generally, for the sports sector. On the one hand, their usability could be, for example, the visualization of different body limbs or even a full-body visualization using real-time inverse kinematics in VR and, on the other hand, in providing feedback to individual users about their movement patterns by collecting positional data of body parts in virtual environments. Therefore, we examined the measurement accuracy of the HTC VIVE Tracker 3.0 during real athletic movements. To cover different typical movement patterns of athletic skills, we tracked the positions of the knee and the ankle during cycling on a bicycle ergometer. While the knee followed an approximately one-dimensional linear motion between two points with high translational accelerations and velocities, the ankle moved almost circularly at a constant radial velocity and acceleration. Furthermore, the influence of various factors (pedaling frequency and positioning of the base stations) was also investigated to generate the prerequisites for adequate measuring. For this purpose, the position data of the VIVE trackers were compared with the gold-standard Vicon. We chose to calculate the spatial distance between the trajectories of the systems at the knee and ankle as well as the radius of the nearly circular ankle motion from the raw data as parameters, allowing for the comparison of the two measurement systems. Since difficulties appeared in synchronizing the VIVE trackers with Vicon as previous studies also reported [25], we corrected the timestamps of the VIVE data based on the minimum and maximum amplitude of each cycle to compare the measured positions at each point in time. As a result of the different acquisition frequencies of the two measurement systems, we interpolated the time-dependent position data to the common frequency of 80 Hz.

Although significant differences occurred concerning the chosen parameters, we saw great potential for using the trackers within future training sessions within virtual environments. It was not surprising that there was a significant deviation determined between the systems for the spatial distance at both the knee and the ankle, since the sample size was large and both technological systems were based on different metrologies. The assumption resulted from the outcome of previous studies’ results, which compared sensors based on other measurement techniques with Vicon. Smaller deviations were found for the Kinect system, for example, (5 mm ± 1.5 mm) [26,27,28]. The measurement accuracy could suffer in terms of its validity and reliability caused by faster movements [29]. This also occurred in the current study, since the differences between the VIVE trackers and Vicon significantly increased at higher movement speeds. Previous studies already evaluated the trackers for their specific purpose, and deviations in the millimeter range were also reported [20,30]. Nevertheless, they emphasized that the trackers did not meet the prerequisites needed for precise measurements for robotic applications. Although the authors recommended the usage of the trackers for indoor and outdoor experiments, it is essential to specify the purpose of their inclusion.

The analysis showed that the pedaling frequency influenced the measurement accuracy of the VIVE trackers much more than the arrangement of the HTC VIVE base stations. Here, it could be shown that the spatial distance between VIVE and Vicon significantly increased with increased movement speed. While the deviations at the knee increased by approximately 5 mm for each frequency level, they increased by as much as approximately 10 mm at the ankle. It could also be seen from the results that the measurement accuracies at low rotational speeds at the knee and ankle were at the same level. With increasing frequency, significantly higher spatial deviations could be observed for the tracker at the ankle compared to Vicon than at the knee. On the one hand, this could possibly be because there was generally less deviation from the trajectory for the rectilinear motion of the knee than for the circular motion of the ankle. On the other hand, the constant change in the direction of the tangential motion on the circular trajectory may have caused larger positional deviations to be measured at the ankle. A smaller, but still significant, influence on the measurement accuracy of the VIVE trackers had the positioning of the base stations. The rectangular arrangement of the base stations recommended by the manufacturer [31] was confirmed as the optimal positioning for both the tracker at the knee (*d*_rect_ = 15.3 mm ± 9.3 mm) and the one at the ankle (*d*_rect_ = 20.8 mm ± 11.6 mm). A kite-shaped arrangement of the base stations with the background of allowing for direct visual contact and reducing the distance of the trackers to three of the four base stations missed the target and led to a lower measurement accuracy of the VIVE trackers at the knee (*d*_kite_ = 20.1 mm ± 16.0 mm) and ankle (*d*_kite_ = 26.6 mm ± 18.5 mm). However, there was a strong interaction effect between the positioning of the base stations and the pedaling frequency. While there were significantly greater deviations for the kite-shaped arrangement at 80 rpm and 160 rpm, its measurement accuracy for the pedaling frequency of 120 rpm was only minimally more accurate than that of the rectangular arrangement. Nevertheless, in summary, the rectangular arrangement provided more accurate results than the kite-shaped one.

To be able to determine the direction of the absolute spatial distances between the position data measured with Vicon and VIVE, the radii of both measurement systems for the movement of the ankle were compared. The inferential statistical analysis showed that the measurement system had a significant and by far the largest effect on the measured radius. Furthermore, the pedaling frequency had a very strong influence on the measurement accuracy, whereby, analogous to the previous results, the deviations also increased with increasing frequency. The positioning of the base stations had no effect on the radius. The descriptive statistics showed that the radii measured with the VIVE trackers were always larger than those measured with Vicon for all conditions. The most obvious reason that the circular motions of VIVE were overestimated was due to their construction and operation. The VIVE trackers’ position data were calculated via the double integration of accelerations measured with the IMU and calibrated in space using the base stations to counteract typical drift. However, due to the double integration, even minimal deviations of the measured accelerations were potentiated in the resulting position. The fact that the deviations became steadily larger with increasing accelerations proved the cause described. Accelerations of more than 600 m/s^2^ occurred for the largest pedaling frequency examined. In combination with the maximum possible recording frequency for the wireless transmission of the VIVE trackers of 90 Hz, the high accelerations led to a delayed reversal of direction due to the double integration, which was why movements that were not linear were overestimated.

Due to the observed deviation between the specified and the actual recording frequency of the VIVE trackers, their time stamps had to be corrected using the data recorded with Vicon. However, such a correction was only possible if a second, parallel running measurement system was available. Accordingly, further investigations should find out why, on the one hand, the actual recording frequency of the VIVE trackers deviated from the set one and, on the other hand, why these deviations were not constant across all trials. A possible cause could be an uneven utilization of the CPU, or the limited processing power of the hardware used, although these met the requirements specified by the manufacturer. Despite the adjustment of the time stamps of VIVE, only an approximation could be achieved. Random deviations, which could not be corrected with the linear regression, sometimes led to significant deviations of the spatial distance between both measurement systems due to the fast movements.

In some cases, very high deflections in the position time series were measured outside the analyzed measurement period, which did not correspond to the cyclic movement of the tracker observed from the outside. This observation was consistent with the dropouts of the VIVE trackers previously found by Verdelet et al. [32], where the trackers in our investigations also needed approximately 3 s to regain their position in space and provide reliable values.

Concerning the practical usage of the trackers within sport-specific virtual training scenarios, we saw a high potential to visualize task-relevant body parts accurately. The minimal shifting to the body position of an already evaluated system such as Vicon was only a few millimeters, and the humans’ perception in virtual environments was manipulatable to a certain degree. Additionally, the trackers’ data could be captured throughout the training session to ensure the individual positional feedback of participants’ body limbs. These data could be used to instruct the participants on their rough movement patterns. However, caution should be exercised when determining precise performance diagnostic parameters, as these results may be subject to error, especially at high velocities and accelerations, due to temporal offset and spatial deviation. One should keep in mind that Vicon includes complex full-body models that calculate the natural joint’s origin, whereas the trackers are only placed on the surface. Therefore, they can first be used in terms of the fundamental placement of, for example, step length, turnings of specific body limbs and the chronological sequence of these.

## 5. Conclusions

In this study, the position data from HTC VIVE Tracker 3.0 at the hip, knee and ankle were recorded while riding a bicycle ergometer at different pedaling frequencies and different base station arrangements. For the analysis of the measurement accuracy, the positional data at the knee and the ankle were compared to the gold-standard Vicon by calculating the spatial distance at each point in time from the raw data. The comparison of the various influencing factors showed that the movement speed represented by the pedaling frequency had the greatest effect on the accuracy of the measured positions. The influence of the base station positioning on the measurement accuracy was smaller, but also significant, with the rectangular arrangement showing the smallest differences to the gold standard. Nevertheless, the quality of the measured values for uniform as well as circular movements was high, even during various faster athletic skills with velocities up to 3.3 m/s and accelerations of more than 600 m/s^2^. Regardless of the influencing factors, the measurement accuracy was sufficient for reconstruction of body parts using inverse kinematics for visualization in VR for adequate movements. In addition, the HTC VIVE trackers were also suitable for measuring the position of, as well as distances between, body points or joints in field tests, where Vicon’s complex and expensive laboratory setup, including calibration, was not practical. For the rapid evaluation of athletic movements, the measurement accuracy was also high enough to be able to draw first conclusions about the quality of the executed technique from the measured body positions and, thus, revealed possibilities for its optimization. However, the VIVE trackers would not be suitable for high-precision biomechanical analyses.

Although, in this study, we investigated both linear and circular movements for different velocities and accelerations typical for a variety of athletic skills, the investigated activity of cycling was characterized by low amplitudes and cyclic movements. Therefore, future studies should confirm the results for such movements with a larger and more complex range of motion. The temporal resolution of the HTC VIVE trackers should also be analyzed in more detail in conjunction with Unity during wireless transmission of the data.

## Figures and Tables

**Figure 1 sensors-23-07371-f001:**
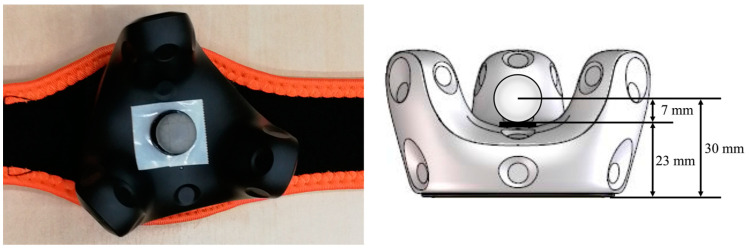
HTC VIVE Tracker 3.0 with attached passive Vicon marker (diameter: 14 mm).

**Figure 2 sensors-23-07371-f002:**
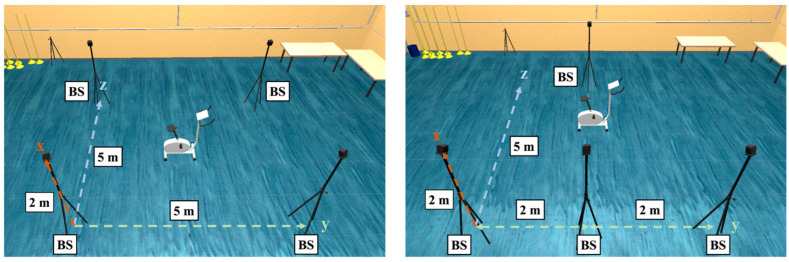
Arrangement of the four base stations (BS) as rectangle (**left**) and kite (**right**).

**Figure 3 sensors-23-07371-f003:**
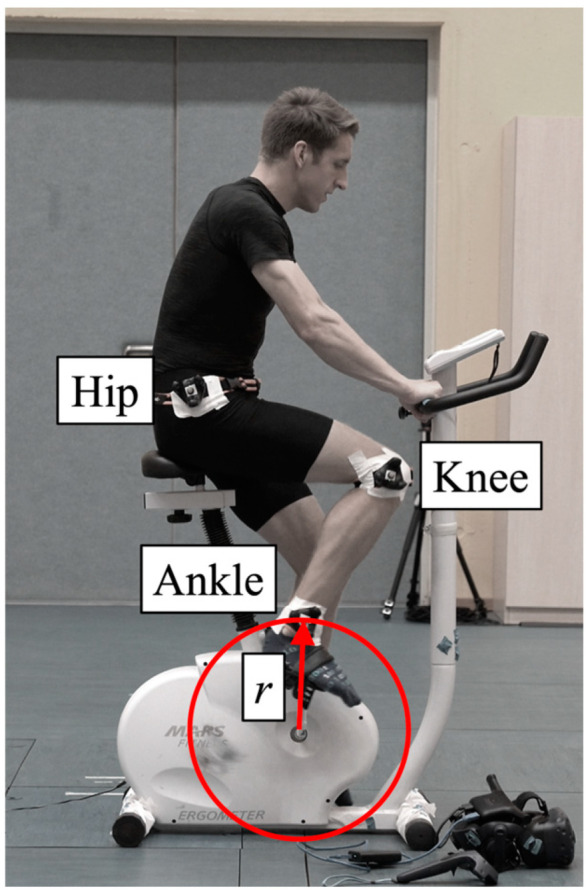
Illustration of the experimental setup and the positions of the three HTC VIVE Tracker 3.0 trackers with attached Vicon markers at the participant’s hip, knee and ankle. In addition to comparing the trajectories at the knee and ankle, the radius *r* of the ankle motion was calculated as a measurement accuracy parameter.

**Figure 4 sensors-23-07371-f004:**
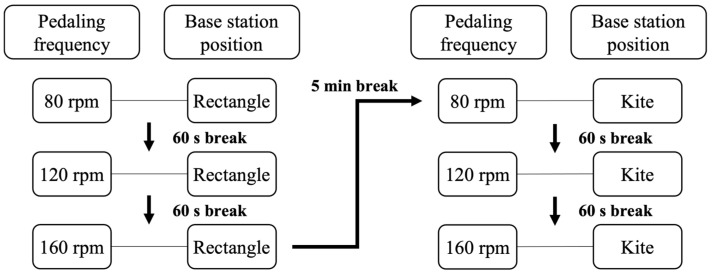
Schematic representation of the six measurement conditions in their chronological order.

**Figure 5 sensors-23-07371-f005:**
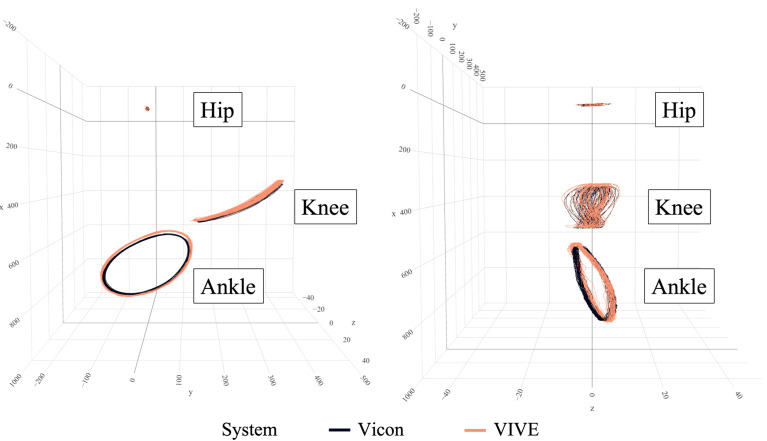
Side (**left**) and rear view (**right**) of the spatially synchronized 3D trajectories of the three trackers and markers (hip, knee and ankle) measured with VIVE and Vicon for the first trial with 80 rpm pedaling frequency and rectangle base station position.

**Figure 6 sensors-23-07371-f006:**
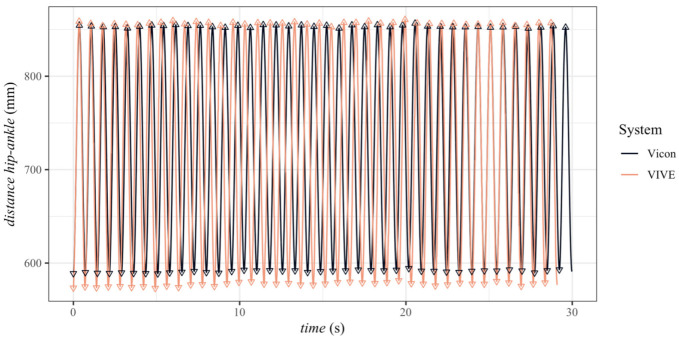
Time series of distances between hip and ankle of the VIVE trackers and Vicon markers with extracted minimum and maximum values for each cycle for the first trial with 80 rpm pedaling frequency and rectangle base station position.

**Figure 7 sensors-23-07371-f007:**
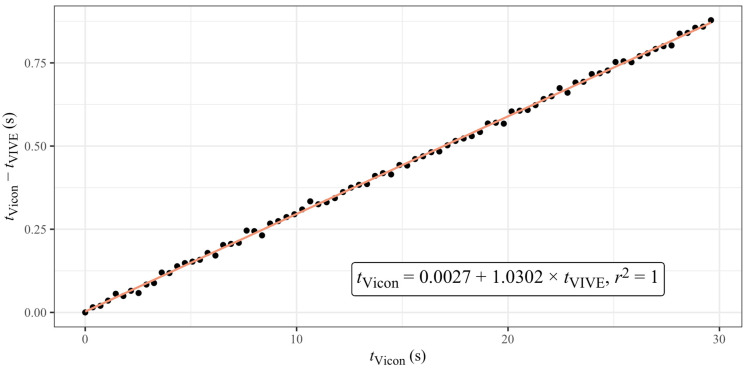
Time difference between VIVE and Vicon of the related extreme values as a function of the time measured with Vicon for the first trial with 80 rpm pedaling frequency and rectangle base station position. From the slope of the linear regression, the actual recording frequency of the VIVE trackers for this trial was 87.4 Hz.

**Figure 8 sensors-23-07371-f008:**
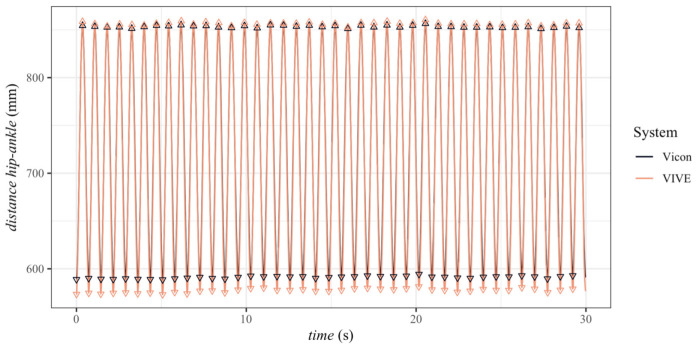
Corrected distance–time series of the VIVE trackers compared to the original values of the Vicon markers with extracted minimum and maximum values for each cycle for the first trial with 80 rpm pedaling frequency and rectangle base station position.

**Figure 9 sensors-23-07371-f009:**
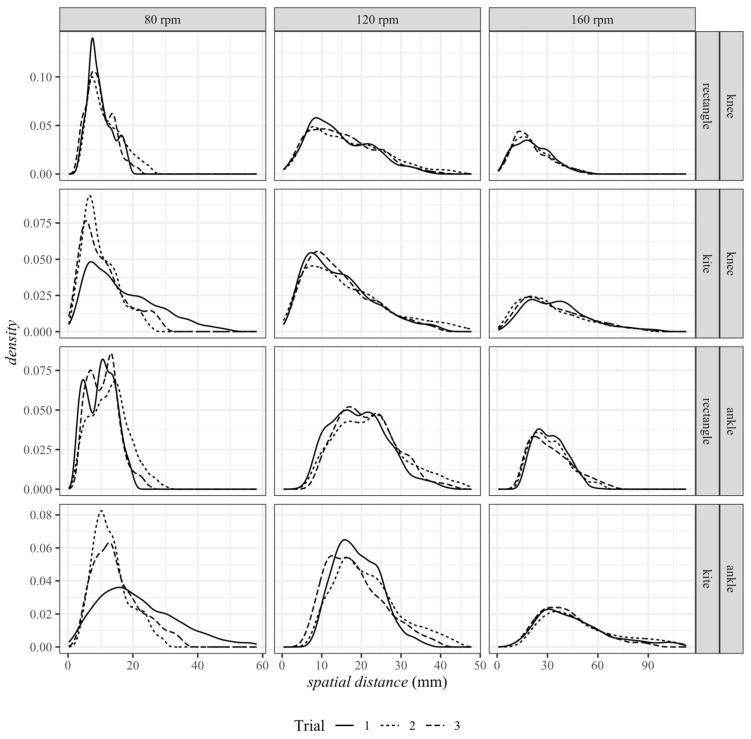
Distributions of the spatial distance (in mm) between VIVE and Vicon for all trials and the factor pedaling frequency (80 rpm, 120 rpm and 160 rpm), tracker position (knee and ankle) and base station position (rectangle and kite).

**Figure 10 sensors-23-07371-f010:**
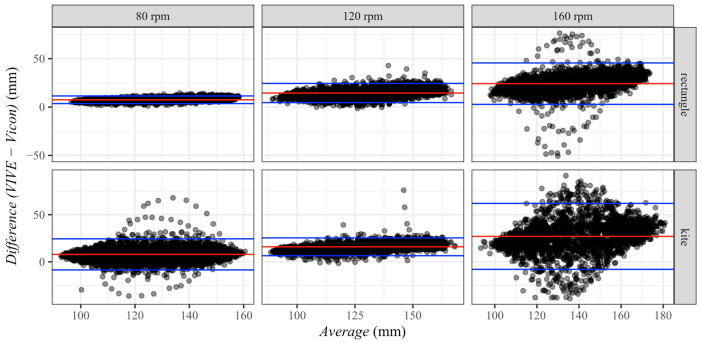
Bland–Altman plots of the ankle motion radius differences and averages between VIVE and Vicon for the factor pedaling frequency (80 rpm, 120 rpm and 160 rpm) and base station position (rectangle and kite). The red line represents the mean difference in each plot, and the blue lines the 95% confidence intervals.

**Table 1 sensors-23-07371-t001:** Inference statistic and descriptive data for the spatial distance (in mm) between VIVE and Vicon at the knee tracker position depending on the factor pedaling frequency (Freq) and base station position (BSP).

Factor		Significance				Effect Size
Two-factor ANOVA with calculated effect size for the spatial distance at the **knee**						
Between-subject effects						
Pedaling frequency		*F* _2, 39,964_ = 7570.814, *p <* 0.001, *η_p_*^2^ = 0.275				Large effect
Base station position		*F* _1, 39,964_ = 2015.136, *p* < 0.001, *η_p_*^2^ = 0.048				Small effect
Interaction effect of the between-subject effects						
BSP × Freq		*F* _2, 39,964_ = 1383.838, *p* < 0.001, *η_p_*^2^ = 0.065				Moderate effect
**BSP**	**Freq**	**mean**	**std**	**n**	***p*-value (post hoc)**	
Rectangle	80 rpm	10.36	4.45	7049	80 rpm–120 rpm	<0.001
	120 rpm	15.51	8.55	7123	120 rpm–160 rpm	<0.001
	160 rpm	20.96	10.98	5896	80 rpm–160 rpm	<0.001
Kite	80 rpm	12.50	8.49	6838	80 rpm–120 rpm	<0.001
	120 rpm	15.02	9.13	6824	120 rpm–160 rpm	<0.001
	160 rpm	33.99	19.13	6240	80 rpm–160 rpm	<0.001

**Table 2 sensors-23-07371-t002:** Inference statistic and descriptive data for the spatial distance (in mm) between VIVE and Vicon at the ankle tracker position depending on the factors pedaling frequency (Freq) and base station position (BSP).

Factor		Significance				Effect Size
Two-factor ANOVA with calculated effect size for the spatial distance at the **ankle**						
Between-subject effects						
Pedaling frequency		*F* _2, 40,490_ = 17,132.906, *p* < 0.001, *η_p_*^2^ = 0.458				Large effect
Base station position		*F* _1, 40,490_ = 2781.516, *p* < 0.001, *η_p_*^2^ = 0.064				Moderate effect
Interaction effect of the between-subject effects						
BSP × Freq		*F* _2, 40,490_ = 1512.541, *p* < 0.001, *η_p_*^2^ = 0.070				Moderate effect
**BSP**	**Freq**	**mean**	**std**	**n**	***p*-value (post hoc)**	
Rectangle	80 rpm	11.33	5.14	7179	80 rpm–120 rpm	<0.001
	120 rpm	21.05	7.72	7131	120 rpm–160 rpm	<0.001
	160 rpm	31.88	11.10	5996	80 rpm–160 rpm	<0.001
Kite	80 rpm	16.67	9.41	6905	80 rpm–120 rpm	<0.001
	120 rpm	19.60	7.21	7028	120 rpm–160 rpm	<0.001
	160 rpm	45.54	20.73	6257	80 rpm–160 rpm	<0.001

**Table 3 sensors-23-07371-t003:** Inference statistic and descriptive data for the ankle motion radius (in mm) between the VIVE and Vicon systems depending on the factor pedaling frequency (Freq) and base station position (BSP).

Factor	Significance	Effect Size
Three-way ANOVA with repeated measurements and calculated effect size for the **ankle** motion radius
Between-subject effects
Pedaling frequency	*F* _2, 14,394_ = 280.690, *p* < 0.001, *η_p_*^2^ = 0.038	Small effect
Base station position	*F* _1, 14,394_ = 1.926, *p* = 0.165, *η_p_*^2^ = 0.000	No effect
Within-subject effects
System	*F* _1, 14,394_ = 40,216.152, *p* < 0.001, *η_p_*^2^ = 0.736	Large effect
Interaction effects of the within-subject effects
System × Freq	*F* _2, 14,394_ = 4136.003, *p* < 0.001, *η_p_*^2^ = 0.365	Large effect
System × BSP	*F* _1, 14,394_ = 88.489, *p* < 0.001, *η_p_*^2^ = 0.006	Small effect
BSP × Freq × System	*F* _2, 14,394_ = 19.655, *p* < 0.001, *η_p_*^2^ = 0.003	Small effect
**BSP**	**Freq**	**System**	**Mean**	**std**	**n**	***p*-value (post hoc)**
Rectangle	80 rpm	VIVE	133.53	19.24	2400	<0.001
		Vicon	125.99	18.09	2400	
	120 rpm	VIVE	138.49	22.65	2400	<0.001
		Vicon	123.97	20.12	2400	
	160 rpm	VIVE	149.06	23.54	2400	<0.001
		Vicon	124.86	19.15	2400	
Kite	80 rpm	VIVE	132.52	19.77	2400	<0.001
		Vicon	124.64	18.67	2400	
	120 rpm	VIVE	138.71	23.21	2400	<0.001
		Vicon	122.79	20.60	2400	
	160 rpm	VIVE	153.53	25.20	2400	<0.001
		Vicon	126.53	20.56	2400

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
