# Peer review of "Measurement Accuracy of the HTC VIVE Tracker 3.0 Compared to Vicon System for Generating Valid Positional Feedback in Virtual Reality"

_sensors, 2023, doi:10.3390/s23177371_

Round 1

Reviewer 1 Report (Previous Reviewer 3)

I consider that the improvements to the work were significant and that it can be published in this way, in general, the methodology and scope of the work were clarified.

Author Response

Thank you very much for your constructive feedback. You helped to improve the paper's quality. 

Yours sincerely, 

Stefan Pastel 

Reviewer 2 Report (Previous Reviewer 2)

My comments have been sufficiently addressed. 

Author Response

Thank you very much for allowing us to re-edit the submitted manuscript. Thank you for your constructive feedback. 

Sincerely yours,

Stefan Pastel 

Reviewer 3 Report (Previous Reviewer 1)

The manuscript and soundness of the manuscript has much improved with aligning the data properly.

I do still find it hard to understand the statistical analysis done. the anova right now is done on the radius of the ankle movement and the length of knee movement?

Personally I would prefer knowing what the deviation between vicon and vive motion tracking is so we can estimate how accurate Vive tracks motion then an anova can be done to see if this accuracy changes significantly (preferably with an assessment of meaningful change) with speed and lighthouse position.

Author Response

Thank you very much for your constructive feedback. We also explained your mentioned points and have referred to the text lines. We hope that the upcoming questions are answered. Thanks a lot for the detailed feedback. You helped to increase the paper's quality. 

This manuscript is a resubmission of an earlier submission. The following is a list of the peer review reports and author responses from that submission.

Round 1

Reviewer 1 Report

This manuscript aims to asses the measurement accuracy of HTC Vive trackers 3.0.

While I am very interested in how the different speeds of movement and the base station setup affects the HTC Vive tracking accuracy the analysis of the data has some fundamental flaws. As the authors state in the discussion they have used a single Vicon marker placed (from fig.1 i assume it is an 0.9cm marker but i could be wrong) on top of a HTC Vive tracker (thickness of about 2cm). Because of the difficulty with the irregularity of unity providing the positional data of the trackers you decided to compare maxima and minima of the angles and distances of the markers without ever estimating the positional error caused by this distance between the HTC Vive tracker position and the Vicon marker position (statically already about 2.5cm) and when you start calculating distances between 2 points in space this easily could already add up to 5cm error. When you start calculating angles with the orientation of the HTC Vive trackers with Vicon markers attached not being constant you introduce triangular errors that become quite unpredictable.

Aditionally, the fact that you compare two very different capture frequencies does not seem right either. However HTC Vive should be goo up to 15.000 degrees/s (90Hz/2.1 (nyquist freq)=42hz which is equal to 15.120 degrees/s).

your english is clear, however the last sentence of the abstract seems unfinished.  on the other hand to make statements about the qualtity.. I assume you wanted to say the measurement accuracy is too low for sports science analysis.

Reviewer 2 Report

This paper evaluated the accuracy of VIVE 3 trackers against a traditional optical motion capture system. This reviewer believes that understanding the accuracy of VIVE 3 trackers is beneficial to the biomechanics community. However, this study suffered for several methodological limitations. In particular, the inability to align the coordinate systems of the motion trackers will contribute to deviations that may be over-estimated. Furthermore, the use of a single participant, and a task that requires limited motion capture volume limits the generalizability of this study.

Introduction:

It would be good to also mention how the VIVE trackers work, and why this technology is appealing compared to other non-maker-based motion capture technologies (markerless motion capture, inertial motion capture). 

Methods:

Can you elaborate whether the VIVE tracker provides orientation information, or just position?

Was there calibration procedure for the VIVE trackers?

Have you considered optimization-based methods to align trackers positionally (e.g. hand-eye calibration, AX=YB)? 

Section 2.3:

Can you not interpolate / resample the VIVE data to a consistent sampling frequency?

Discussion:

Why is only one participant used?

Reviewer 3 Report

The article presents the comparison of two systems for generating valid positional feedback in a particular sport activity. In my opinion, the subject does not have a research scope, it is an important technical report to know the systems, but the contribution of the systems is not of the authors and no methods are presented that contribute to the adjustment or that demonstrate a significant contribution.

I consider that it is not possible to identify the contribution of the authors, that an experiment was carried out that is even limited to say that it is really a comparison, and that in an academic work for this type of journal another type of contribution is expected.

In my opinion, the paper represents a good technical exercise that can be useful for the development of interpolation or measurement models, but in this form, the authors do not present alternatives of their own and the comparisons in my opinion are not sufficient.